Neglected Tropical Diseases

# People with heart failure, sarcopenia and chagas disease: A systematic review and meta-analysis

**Melissa Orlandin Premaor** [ID]*, **Suellen de Azevedo Mendes,**
**Gustavo Henrique Silva Ambrósio Vieira, Camila Diniz Braga, Aron Fonseca Santos,**
**Diego Silva Assaf Ferreira, Bruno Cesar Fernandes Araujo,**
**Maria do Carmo Pereira Nunes, Luis Felipe Jose Ravic de Miranda**

Department of Clinical Medicine, Medical School, Federal University of Minas Gerais, Belo Horizonte, Brazil

\* premaor@medicina.ufmg.br

## Abstract

### Background

This systematic review aims to estimate the prevalence of sarcopenia in people with heart failure (HF) and assess whether there is a difference in sarcopenia prevalence based on the underlying cause of HF, specifically comparing HF due to Chagas (HF-C) disease with HF from other causes (HF-NC).

### Methods

A systematic review of randomized controlled trials, cohort studies, cross-sectional studies, and case-control studies was carried out. Individuals with HF over 18 years who had their frequency of sarcopenia evaluated were included.

### Results

Overall, 3347 studies were found, of which 199 had their full texts evaluated. A total of 25 records were identified. The prevalence of sarcopenia, as defined by EWG-SOP2 criteria, was 23.27% (95%CI 15.4, 33.6). Further, there were differences in sarcopenia prevalence according to the definition used in the studies (p-value = 0.0002). The odds ratio of sarcopenia in people with HF was 2.3 (95%CI 1.1, 4.8). Two studies specifically evaluated sarcopenia in HF-C, reporting a prevalence of 24.2% (95%CI 12.6, 41.5). The odds ratio for sarcopenia in HF-C was 1.93 (95% CI 0.40, 9.30) compared to HF-NC.

### Conclusions

Patients with HF have an increased risk of developing sarcopenia. However, the prevalence of sarcopenia varies according to the definition used. Furthermore, our

**Data availability statement:** The Data Availability Statement DOI is https://doi.org/10.6084/m9.figshare.28477061 and the link to access it is https://doi.org/10.6084/m9.figshare.28477061.v1.

**Funding:** This work was supported by the the Fundação de Amparo à Pesquisa do Estado de Minas Gerais (EDITAL 100/2024-BOLSAS DE PRODUTIVIDADE EM PESQUISA NÍVEL E - CHAMADA CNPQ Nº 09/2023" - ESTUDO DA SARCOPENIA E FRAGILIDADE ÓSSEA EM PESSOAS QUE VIVEM COM INSUFICIÊNCIA CARDÍACA CAUSADA POR DOENÇA DE CHAGAS - APQ-06623-24" Ref. Fundep nº 31960 to MOP). The funders had no role in study design, data collection and analysis, decision to publish, or preparation of the manuscript.

**Competing interests:** The authors have declared that no competing interests exist.

findings highlight the hypothesis that the risk of sarcopenia in HF-C might be higher than HF-NC.

## Author summary

A systematic review of randomized controlled trials, cohort studies, cross-sectional studies, and case-control studies was carried out to estimate the prevalence of sarcopenia in people with heart failure (HF) and assess whether there is a difference in sarcopenia prevalence based on the underlying cause of HF, specifically comparing HF due to Chagas (HF-C) disease with HF from other causes (HF-NC). Overall, 3347 studies were found, of which 199 had their full texts evaluated. A total of 25 records were identified. There were differences in sarcopenia prevalence according to the definition used in the studies (p-value=0.0002). The odds ratio of sarcopenia in people with HF was 2.3 (95%CI 1.1, 4.8). Two studies specifically evaluated sarcopenia in HF-C, reporting a prevalence of 24.2% (95%CI 12.6, 41.5). The odds ratio for sarcopenia in HF-C was 1.93 (95% CI 0.40, 9.30) compared to HF-NC. Our findings suggest the hypothesis that the risk of sarcopenia in HF-C might be higher than HF-NC.

## Introduction

Heart failure (HF) is one of the most prevalent syndromes worldwide, imposing a significant health burden [1–3]. Coronary heart disease and ischemic cardiomyopathy are the leading causes of HF [3]. Nonetheless, Chagas disease represents a substantial proportion of this syndrome [3,4]. According to the World Health Organization (WHO), approximately 75 million people worldwide are at risk of infection by *Trypanosoma cruzi*, and around 7 million people are infected [5]. Among those infected, approximately 42.7% develop chronic cardiomyopathy with arrhythmias and heart failure [3,4,6].

Currently, HF has been seen as a syndrome that goes far beyond cardiac manifestations [1,7]. Low muscle mass, low muscle strength, and sarcopenia have been described in people with HF [7,8]. The reported prevalence of sarcopenia in HF has been between 10 and 69% [7,8]. Most studies describe that decreased muscle mass and sarcopenia usually precede cardiac cachexia [9,10]. In turn, sarcopenia seems to worsen the prognosis of HF [7,8]. Moreover, several pathological mechanisms present in HF are common to both pathologies: chronic inflammation, oxidative stress, hormonal changes, poor nutrition, and physical inactivity [7].

Patients with HF due to Chagas disease may be especially prone to sarcopenia. Chagas disease results in a persistent low-grade parasitic infection, leading to chronic inflammation and an abnormal immune response [2–4]. Patients with HF due to Chagas disease have been described as experiencing increased oxidative stress, increased cytokines, and increased inflammatory proteins [2,3]. Chronic inflammation

and increased cytokines are common mechanisms in sarcopenia. Nonetheless, Chagas disease has been a neglected disease [3,4], and it remains unclear whether the prevalence of sarcopenia differs between patients with HF due to Chagas disease and those with HF from other etiologies.

In this systematic review and meta-analysis, we aimed to assess the prevalence of sarcopenia in patients with HF. We also compared its prevalence between those with Chagas disease and those with other HF causes.

## Materials and methods

This study was designed as a systematic review and meta-analysis. It has followed the PRISMA- Preferred Reporting Items for Systematic Reviews and Meta-analyses (PRISMA) Guidelines [11]. The review protocol was registered at the University of York database (PROSPERO) under the number CRD42023401386.

### Eligibility criteria

Randomized controlled trials, cohort studies, cross-sectional, and case-control studies that included men and women over 18 years of age, with HF of any cause (HF-NC) and HF caused by Chagas disease (HF-C), and that had their frequency of sarcopenia evaluated, were eligible to this systematic review. There were no restrictions on the language and year of publication of the studies. Studies conducted with men and women under 18 years or without HF, in vivo, in vitro studies, other study designs, and studies that have evaluated specific subgroups of HF patients (e.g., studies that evaluate hospitalized patients) were excluded. We decided to exclude inpatients due to the chance that they might have some degree of acute sarcopenia as suggested by Cruz-Jentoft et al [12].

### Data sources and search strategy

The search for the studies was performed at the Medical Literature Analysis and Retrieval System Online (MEDLINE, PubMed tool), the Excerpta Medica dataBASE (EMBASE), and the Regional Library of Medicine (BIREME, BVS tool). Moreover, some studies were searched based on the reference lists of the included articles. The previously defined terms for each database are displayed in the Table A in S1 Text. The last search was performed on July 9th, 2024.

### Study selection and strategy for the identification of eligible studiesx

The references retrieved in the search strategy were imported into an Excel spreadsheet, and the duplicates were removed. The protocol members independently performed the study selection in duplicate. The screening was based on the titles and abstracts of the articles. The full texts were sought for all studies that appeared to meet the eligibility criteria and/or that could not be ruled out in the screening. The study's eligibility was double-checked for all selected items. If there was disagreement between the above protocol members regarding the identification, eligibility, and inclusion of items, they were rechecked by an adjudicator protocol member.

### Data extraction

The protocol members extracted the data in duplicate and independently. The agreement between the extractors was 90%. If there was disagreement between the protocol members regarding the extraction, they were rechecked by an adjudicator member. The data extracted for this study are displayed in the Table B in S1 Text. The data shown as figures were extracted with the WebPlotDigitizer version 4.6 (Ankit Rohatgi; https://www.gnu.org/licenses/agpl-3.0.html). If the data could not be extracted from the full text of a study, authors were contacted by email at least three times. Two authors have responded. Further, whenever different articles from the same cohort were obtained, all relevant articles were included in the review to ensure comprehensive data extraction.

## Quality and risk of bias in individual studies

For the cohort, case-control, and cross-sectional studies, the bias risk was assessed using the Newcastle-Ottawa scale (NOS) for cohort and case-control studies and the modified NOS for cross-sectional studies, respectively [13]. For the RCT, NOS for case-control was used due to the necessity of the evaluation of sampling and comparability for the adequate interpretation of the frequency of sarcopenia. NOS addressed the selection, comparability, and outcome of the studies. In this semi-quantitative method, nine stars represent the maximum score for a study. We considered a study of good quality when it scored seven or more stars. The differences in the scoring were resolved by consensus between the senior protocol members. The study data are publicly available and can be found at Figshare (https://doi.org/10.6084/m9.figshare.28477061.v1) [14].

## Data synthesis and statistical analysis

The frequency of sarcopenia was combined using random models. Only one study of the same cohort was included per meta-analysis. The study with the larger number of participants was chosen. A combined confidence interval of frequency and 95% CI was calculated. The inverse variance method, the DerSimonian-Laird estimator for tau^2, was used for all pooled analyses. The statistical heterogeneity among studies was assessed using Cochran's Q test and the inconsistency I2 test. We performed an additional analysis with a fixed-effects model to qualitatively evaluate differences in point estimates provided by models with random and fixed effects. To explain heterogeneity and potential effect modifiers, we performed meta-regression for age, study year, BMI, gender, NYHA class, ejection fraction, study design, method of reporting muscle mass, technique of lean mass evaluation, study location, and study quality using a mixed-effects model (with the explaining variable as fixed).

Furthermore, we conducted a sensitivity analysis to evaluate the influence of individual studies on the heterogeneity, repeating the meta-analysis with each study detected once (live-one-out analysis). The subgroup analysis was carried out for sarcopenia criteria using the Mantel-Haenszel method. For the sarcopenia criteria meta-analysis, all studies that conveyed the necessary information were included. We also assessed evidence of publication bias through a qualitative inspection of the funnel plot and the Begg test as a statistical parameter for testing funnel plot asymmetry. All the analyses were made using the software R [R version 3.2.4, 2016, The R Foundation for Statistical Computing, Platform: x86_64-apple-darwin13.4.0 (64-bit)] and RStudio [RStudio Team (2015). RStudio: Integrated Development for R. RStudio, Inc., Boston, MA URL http://www.rstudio.com/].

The quality of the evidence was evaluated with the GRADE procedure [15].

# Results

## Eligible studies

The overview of the selection process is displayed in Fig 1. Overall, 3345 studies were retrieved from different databases: EMBASE (n = 1666), MEDLINE (PUBMED) (n = 622), BIREME (n = 1057). After removing the duplicated studies, 2143 were excluded from the title and abstract triage. Furthermore, 197 studies underwent a detailed full-text review. Besides, 02 studies were identified from the reference lists of the evaluated records. Of these, 174 were excluded due to a lack of study factor, lack of outcome, study design (in vivo studies, uncontrolled clinical trial), all subjects included had the outcome, all subjects excluded were inpatients, special populations (all ventricular assist device, all acute heart failure, and all aortic aneurysm), and redundant publication. A total of 25 studies evaluated the frequency of sarcopenia in people with heart failure. Three studies had a non-heart failure control group.

## Study characteristics

The characteristics of the studies included in this systematic review are described in Table 1. Overall, 25 papers with a total of 2,507 people with HF were included. The conference paper by Cvjetan et al. [16] is an update from the

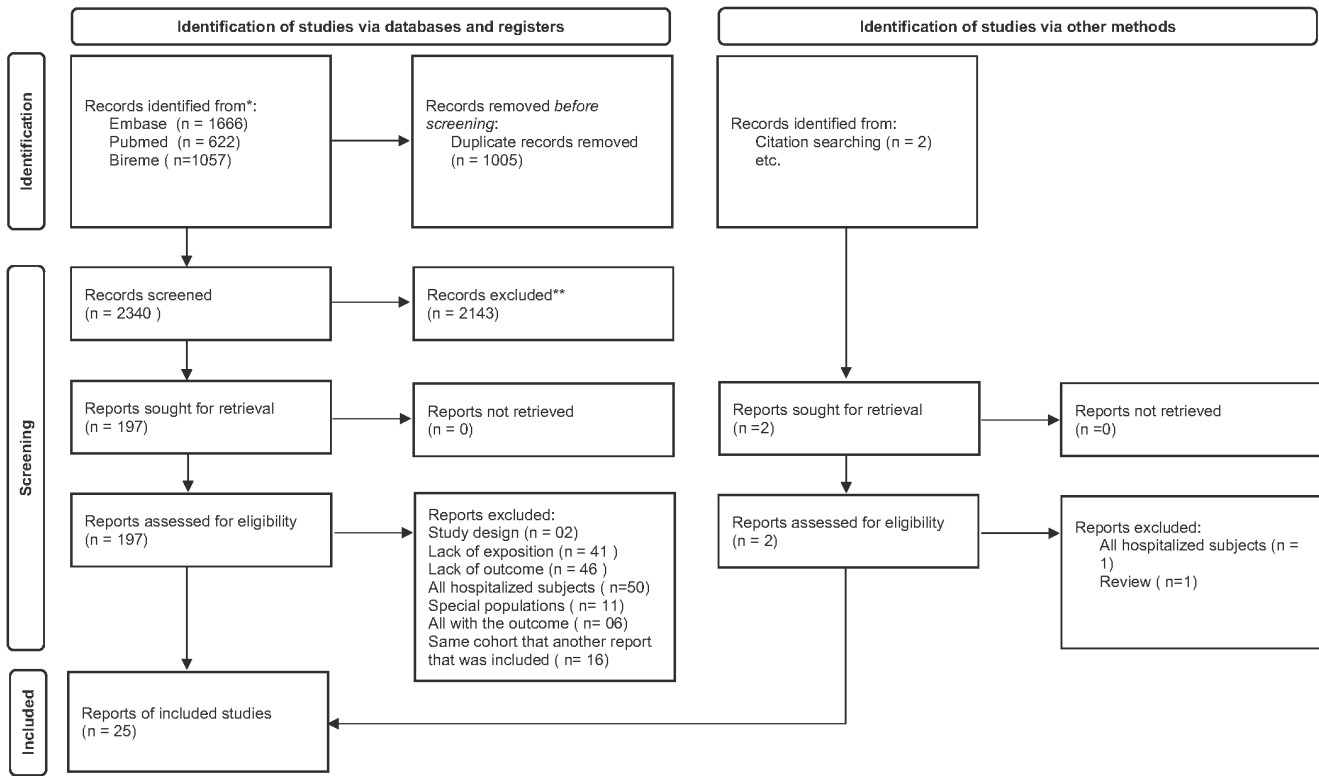

PRISMA 2020 flow diagram for new systematic reviews which included searches of databases, registers and other sources

From: Page MJ, McKenzie JE, Bossuyt PM, Boutron I, Hoffmann TC, Mulrow CD, et al. The PRISMA 2020 statement: an updated guideline for reporting systematic reviews. BMJ 2021;372:n71. doi: 10.1136/bmj.n71.

**Fig 1. PRISMA 2020 flow diagram of the studies included in the review.**

conference paper by Loncar et al. [17], However, both are kept in the systematic review because they provide complementary information. Likewise, the studies by Fonseca et al. [18], Fonseca et al. [19,20], Emmani et al. [21], and dos Santos et al. [22] included subjects from the same cohort. However, they were all kept in the systematic review because they conveyed different information. The most common design was cross-sectional (n = 13). There were six cohorts, one clinical trial, and three case-control studies. Most of the selected studies recruited subjects by convenience, while 17 studies recruited subjects from a tertiary care outpatient clinic. The more frequent study sites were Brazil and Japan. The mean age of the included subjects varied from 37.3 to 80.0 years. Moreover, the mean BMI varied from 23.3 to 29.9 kg/m2. Two studies (three publications) described the frequency of sarcopenia in people with HF due to Chagas disease (HF-C).

The muscle mass was evaluated by dual-energy X-ray absorptiometry in 14 studies (Table 2). Further, six studies have estimated the muscle mass using bioimpedance, and two have estimated it using anthropometric measurements (Table 2). One study has estimated muscle mass using the Forbes formula (Table 2) [23]. As the Forbes formula estimates muscle mass based on serum creatinine levels, which has some controversy, this latter study was excluded from the meta-analysis. The muscle mass was reported as appendicular skeletal muscle mass index [ASMI, appendicular skeletal muscle mass/height2), n = 16], skeletal muscle mass index [(SMI, skeletal muscle mass/height2), n = 3], calf and the mid-upper arm circumferences (n = 2) and free fat mass index [calculated by the Forbes formula, n = 1; Table 2. One study has also calculated the muscle mass index = muscle mass/BMI [24].

PLOS Neglected Tropical Diseases

**Table 1. Characteristics of the studies included in the systematic review.**

| Author | Year | Study design | Site | Settings | Number of participants | Number of PLHF | Age HF (yrs) | Female % | BMI HF | Ejection fraction | NYHA III/IV | Number of participants with Chagas | Quality of the study |
|---|---|---|---|---|---|---|---|---|---|---|---|---|---|
| Formiga et al | 2024 | Cohort | Spain | Populational | 1420 | 226 | 80.0 (5.0) | 54.4 | | | | No description | 8 |
| Bieger et al. | 2023 | Cohort | Brazil | Tertiary-care | 106 | 106 | 69 (7.0) | 33 | 26.9 (4.5) | 35.9 (4.5) | 17.9 | No description | 7 |
| Valdiviesso et al. | 2022 | Cross-sectional | Portugal | Tertiary-care | 136 | 136 | 59.7 | 33.8 | 29.2 (4.7) | 38.0 (16.5) | 16.2 | No description | 9 |
| Karim et al. | 2022 | RCT | Pakistan | Tertiary-care | 92 | 92 | 66.4 (5.6) | 0 | 23.3 (3.2) | | | No description | 8 |
| Cvjetan et al. | 2022 | Cohort | Serbia | Tertiary-care | 146 | 146 | 68.9 (7.0) | 0 | 24.3 | 29.0 (8.0) | | No description | 7 |
| Karim et al. | 2022 | Case-control | Pakistan | Tertiary-care | 195 | 67 | 68.2 (4.7) | 0 | 24.8 (2.8) | | | No description | 8 |
| Fonseca et al. | 2022 | Cross-sectional | Brazil | Tertiary-care | 169 | 169 | | 0 | 25.6 (3.8) | | | No description | 8 |
| Pinijmung et al | 2022 | Cross-sectional | Thailand | Tertiary-care | 152 | 152 | 58.5 (11.8) | 29.6 | 26.5(4.9) | 37.3 (13.2) | 16.0 | No description | 7 |
| Fonseca et al. | 2020 | Cross-sectional* nested in two RCT | Brazil and Germany | Tertiary-care | 64 | 64 | 63.0 (14.0) | 0 | 26.2 (5.1) | 27.6 (8.0) | | 22 | 8 |
| Fonseca et al. | 2020 | Cross-sectional* nested in one RCT | Brazil | Tertiary-care | 168 | 168 | 58 | 0 | 27.5 (4.2) | 27 | 57 | 40 | 8 |
| Telfer et al. | 2020 | Cross-sectional | USA | No description | 28 | 11 | 64 | 9.1 | 28 | | | No description | 7 |
| Sun et al. | 2020 | Cross-sectional | China | Secondary-care | 122 | 122 | | | | | | No description | 7 |
| Canteri et al. | 2019 | Case-control | Brazil | Echocardiography Service | 222 | 79 | 65.6 (13.0) | 41.8 | 26.9 (4.0) | | | 11 | 8 |
| Loncar et al. | 2019 | Cohort | Serbia | Tertiary-care | 73 | 73 | 68.9 (7.0) | 28.8 | 23.5 (3.1) | 29.0 (8.0) | | No description | 7 |
| Nozaki et al. | 2019 | Cohort | Japan | Tertiary-care | 191 | 191 | 73.3 (7.3) | | | | | No description | 7 |
| Fonseca et al. | 2019 | Cross-sectional | Brazil | Tertiary-care | 116 | 116 | 55.9 (9.0) | 0 | 25.5 (4.5) | 28.0 (8.0) | 30.2 | No description | 8 |
| Gulyaev et al. | 2019 | Cross-sectional | Russia | Tertiary-care | 63 | 63 | 77.2 (7.7) | 50.8 | 28.5 (5.8) | | | No description | 7 |
| Emami et al. | 2019 | Cross-sectional | Germany | Tertiary-care | 207 | 207 | 67.3 (10.1) | 0 | 28.8 (5.0) | 36.9 (12.5) | | No description | 8 |
| Dos Santos et al. | 2019 | Cross-sectional | Brazil | Tertiary-care | 113 | 113 | 55.0 (9.0) | 0 | | | | No description | 9 |
| Watanabe et al. | 2017 | Cross-sectional | Japan | No description | | | | | | | | No description | 0 |
| Hajahmadi et al. | 2017 | Cross-sectional | Iran | Tertiary-care | 55 | 55 | 37.3 (10.1) | 58 | 24.3 (3.7) | 21.4 (9.4) | 12.7 | No description | 7 |
| Molinero-Abad et al. | 2015 | Cohort | Chile | Tertiary-care | 103 | 103 | 82.5 | 48.5 | | | | No description | 6 |
| Obata et al. | 2015 | Cross-sectional | Japan | Tertiary-care | 49 | 49 | 68.0 (13.0) | 33.0 | | 50.0 (18.0) | | No description | 7 |
| Haykowsky et al | 2013 | Case-control | EUA | Tertiary-care | 60 | 60 | 69.8 (7.3) | 68 | 29.9 (4.3) | 65.0 (7.0) | | No description | 8 |
| Landi et al | 2013 | Cohort | Italy | Populational | 197 | 12 | | | | | | | 9 |

**Table 2. Description of the methods used to evaluate the studies outcomes.**

| Author | Year | Sarcopenia (number) | Sarcopenia criterion | Lean mass criterion | Lean mass techinque | Stength criteeion | Strengh techinique | Chagas Sarcope-nia (number) | Comorbidities |
|---|---|---|---|---|---|---|---|---|---|
| Formiga et al. | 2023 | 26 | EWGOSP 2 | ASMI | Bioimpedance | EWGOSP 2 | Handgrip | NI | |
| Bieger et al. | 2023 | 25 | EWGOSP 2 | ASMI | Bioimpedance | EWGOSP 2 | Handgrip Five-Times-Sit-to-Stand Test | NI | HBP 69.4% DM 36.8% |
| Valdiviesso et al. | 2022 | 25 | EWGOSP 2 | calf and mid upper arm circumference | anthropometric measurements | EWGOSP 2 | Handgrip Gait speed | NI | DM 28.6% |
| Karim et al. | 2022 | 32 | EWGOSP 2 | ASMI | Bioimpedance | EWGOSP 2 | Handgrip Gait speed | NI | |
| Cvjetan et al.* | 2022 | 26 | EWGOSP 2 | ASMI | DXA | EWGOSP 2 | Gait speed | NI | |
| Karim et al. | 2022 | 37 | EWGOSP 2 | ASMI | Bioimpedance | EWGOSP 2 | Handgrip | NI | |
| Fonseca et al. | 2022 | 29 | EWGOSP 2 | ASMI | DXA | EWGOSP 2 | Handgrip | NI | |
| Pinijmung et al | 2022 | 29 | AWGS | SMI | Bioimpedance | AWGS | Handgrip Gait speed | NI | |
| Fonseca et al. | 2020 | 15 | EWGOSP 2 | ASMI | DXA | EWGOSP 2 | Handgrip | 5 | |
| Fonseca et al. | 2020 | 66 | Baumgartner | ASM | DXA | EWGOSP 1 | Handgrip | 14 | |
| Telfer et al. | 2020 | 2 | Baumgartner | ASM | DXA | None | | NI | |
| Sun et al. | 2020 | 75 | AWGS | ASMI | DXA | AWGS | Handgrip Gait speed | NI | |
| Canteri et al. | 2019 | 8 | FHNI | ASM/BMI | DXA | FHNI | Handgrip | NI | HBP 78.5% DM 30.4% |
| Nozaki et al. | 2019 | 20 | AWGS | SMI | Bioimpedance | AWGS | Handgrip Gait speed | NI | |
| Fonseca et al. | 2019 | 33 | EWGOSP 1 | SMI | DXA | EWGOSP 1 | Handgrip | NI | |
| Gulyaev et al. | 2019 | 31 | FHNI | ASMI | DXA | None | | NI | |
| Emami et al. | 2019 | 30 | Baumgartner | ASMI | DXA | None | | NI | HBP 59.4% DM 38.2% |
| Dos Santos et al. | 2019 | 75 | Newman | ASMI | DXA | None | | NI | |
| Watanabe et al. | 2017 | | No description | FFMI | Forbes formula | None | | NI | |
| Hajahmadi et al. | 2017 | 16 | Baumgartner | ASMI | DXA | None | | NI | |
| Molinero-Abad | 2015 | 30 | Baumgartner | ASMI | DXA | None | | NI | |
| Obata et al. | 2015 | 17 | EWGOSP 1 | ASMI | DXA | EWGOSP 1 | Handgrip | NI | |
| Haykowsky et al | 2013 | 25 | Newman | ASMI | DXA | None | | NI | |
| Landi et al | 2013 | 5 | EWGOSP 1 | Mid arm circunference | anthropometric measurements | EWGOSP 1 | Handgrip Gait speed | NI | |

EWGSOP2 = sarcopenia definition proposed by European Working Group on Sarcopenia in Older People in 2018

AWGS = sarcopenia definition proposed by Asian Working Group for Sarcopenia in 2014

Baumgartner = sarcopenia definition proposed by Baumgartner et al. in 1998

EWGSOP1 = sarcopenia definition proposed by European Working Group on Sarcopenia in Older People in 2010

FHNI = sarcopenia definition proposed by Foundation for the National Institutes of Health in 2014

Newman = sarcopenia definition proposed by Newman et al. in 2003

DXA = dual-energy X-ray absorptiometry

*The Cyvetan et al study was an updated of the Loncar et al study.

The definitions used to categorize sarcopenia are displayed in Table 2. The most used sarcopenia definition was the definition proposed by the European Working Group on Sarcopenia in Older People (EWGSOP2) in 2018 (Table 2; n = 8) [12]. The second most frequently used definitions were the earlier version of the definition proposed by European Working Group on Sarcopenia in Older People (EWGSOP1) in 2010 [25] (Table 2, n = 3) and the definition

proposed by Baumgartner et al. in 1998 (Baumgartner, Table 2, n = 4) [26]. Further, three studies have used the definition proposed by the Asian Working Group for Sarcopenia (AWGS) in 2014 [27]; 2 studies used the sarcopenia definition proposed by Newman in 2003 [28]; and two studies have used the Foundation for the National Institutes of Health (FNIH, 2014) definition [29].

## Quality of the study

Table 1 summarizes the quality of the studies. One study, published as a conference abstract, presented an unknown bias risk [23]. Nonetheless, all other studies but one (score = 6) were considered of good quality based on the NOS score above or equal to 7. The detailed quality assessment is shown in Table C in S1 Text. The GRADE is displayed in Table F in S1 Text.

## Sarcopenia

The pooled analysis for sarcopenia included 21 studies. The prevalence of sarcopenia in HF patients was 25.68% (95%CI 19.02, 33.70), Fig 2; ⊕⊕⊕○. However, the frequency of sarcopenia assessed by the different criteria has shown a significant difference, p-value = 0.01; Fig A in S1 Text. The lowest prevalence was evaluated by the Baumgartner operational definition [19.79% (95%CI 12.49, 29.91), Fig A in S1 Text], while the highest was the frequency evaluated by the Newman operational definition [55.01% (95%CI 37.55, 71.31), Fig A in S1 Text]. The prevalence of sarcopenia evaluated by the EWGSOP2 definition was 23.27% (95%CI 15.39, 33.58), Fig A in S1 Text.

We carried out the live-one-out analysis (Table D in S1 Text); nonetheless, the pooled results were not affected by this procedure, and no single article could explain the source of the heterogeneity. Further, the p-value for the publication bias evaluated by the Begg test was 0.722. The funnel plot is displayed in Fig B in S1 Text. Although it exhibits symmetry, most studies have a large effect size.

Four studies compared the frequency of sarcopenia between people with HF and people without HF. The odds ratio (OR) of sarcopenia in PHF was 2.3 (95%CI 1.1, 4.8; p-value = 0.02; $I^2$ = 77.2%), ⊕⊕○○. These data are displayed in Fig 3. The p-value for the publication bias evaluated by the Begg test was 0.398. The funnel plot is displayed in Fig C in S1 Text. The study by Formiga et al. [30] appears to explain all the heterogeneity in the live-one-out analysis.

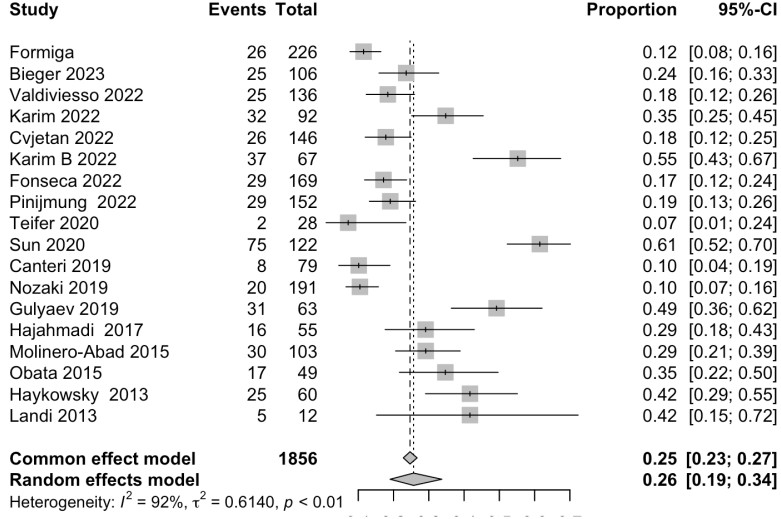

**Fig 2. Forest plot of the frequency of Sarcopenia in people living with heart failure.**

Furthermore, excluding the Karim et al. [31] study reduced the heterogeneity by 23.6%. The live-one-out analysis is shown in Table E in S1 Text.

The results for the meta-regression of the prevalence of sarcopenia in people with HF are shown in Table 3. There was no association between age, study year, BMI, gender, NYHA class, ejection fraction, study design, method of reporting muscle mass, technique of lean mass evaluation, study location, study design, study quality, and the frequency of sarcopenia. Nonetheless, the NYHA class and ejection fraction explained the heterogeneity, at least in part.

### Sarcopenia in patients with heart failure caused by Chagas disease

Fonseca et al. carried out a cross-sectional study nested in two previous cohorts (the SICA-HF study and the TESTO-HF [19]. They evaluated a sample of 64 men with HF, of whom 22 were due to Chagas disease. They found a frequency of sarcopenia of 22.7% according to the EWGSOP2 definition. The researchers also reported the frequency of sarcopenia based on the Baumgartner operational definition in the TESTO-HF study [20]. They included 128 men with HF in this analysis, of whom 40 had Chagas disease. The frequency of sarcopenia in the HF-C was 21.2%, and there were no differences in the frequency of sarcopenia in the HF-C and HF-NC. Canteri et al. included 11 HF-C in their study; the frequency of sarcopenia evaluated by the FNIH criteria was 27.3% [24].

The pooled frequency of sarcopenia in HF-C was 24.2% (95%CI 12.6, 41.5; heterogeneity 0.0%; p = 0.77). Further, the pooled OR of Sarcopenia in HF-C when compared with HF-NC was 1.93 (95% CI 0.40, 9.30; heterogeneity 59.1%; p = 0.12).

## Discussion

In this systematic review, we found that approximately a quarter of people with HF had sarcopenia when it was defined according to the EWGSOP2 definition. However, the frequency of sarcopenia varies according to the operational definition used. Despite this variation, we found that people with HF had twice the chance of having sarcopenia. Although we did not find enough studies to assess the risk of sarcopenia in patients with HF due to Chagas disease, our results suggest that there might be an association between these two pathologies and that further studies are needed to clarify this issue.

The variation in the prevalence of sarcopenia according to the operational criterion used is also present in the general population. Its frequency is described in older adults as between 5 and 38% [32–34]. This variation is expected since the initial criteria (Baumgartner and Newman) did not include muscle strength and function. Furthermore, the different criteria employ different cutoff points for muscle mass, strength, and function [34,35]. In the meta-analysis conducted by Papadopoulou et al., the frequency of sarcopenia in older people, assessed by the EWGSOP1 criterion, was 10% (95%CI 7.0%,

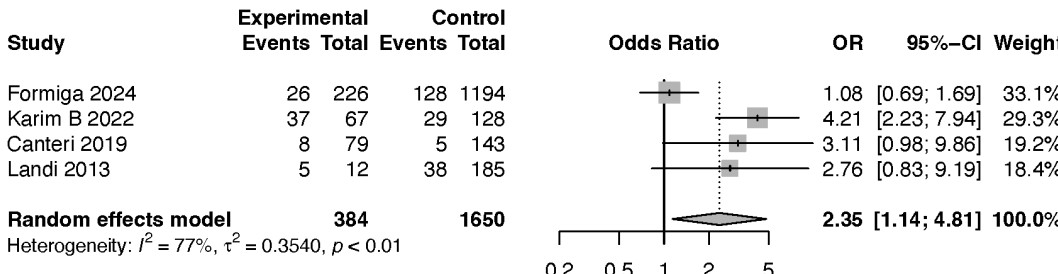

**Fig 3. Forest plot of odds ratio of sarcopenia in people living with heart failure compared with people without this condition.**

**Table 3. Meta-regression of the meta-analysis of the frequency of sarcopenia in people living with heart failure.**

| Variables | No of studies | Change in the Frequency | (95%-CI) | p-value | I², % |
|---|---|---|---|---|---|
| Age | 18 | -0.52 | -0.04; 0.03 | 0.773 | 91.85% |
| BMI | 15 | 0.28 | -17.72; 18.28 | 0.976 | 89.16% |
| Gender (female) | 18 | 0.15 | -1.86; 1.57 | 0.866 | 92.03% |
| NYHA III/IV | 5 | 2.39 | -0.84; 0.05 | 0.148 | 0.0% |
| Ejection Fraction | 9 | 2.00 | -0.17; 4.16 | 0.070 | 62.96% |
| Method of reporting muscle mass (ALMI) | 20 | -23.63 | -84.68; 37.42 | 0.448 | 92.29% |
| Technique of lean mass evaluation | 20 | -19.53 | -79.12; 39.27 | 0.509 | 92.35% |
| Study site | 21 | 18.13 | -11.95; 48.21 | 0.238 | 91.88% |
| Study year | 21 | -8.36 | -20.21; 3.48 | 0.166 | 91.85% |
| Study design | 21 | 11.69 | -29.78; 53.16 | 0.581 | 92.34% |
| Quality of the study | 21 | 16.22 | -31.83; 64.24 | 0.508 | 92.34% |

13.0%), while the prevalences assessed by the AGWS1 and IWGS criteria were, respectively, 11% (95%CI 8.0%, 14.0%) and 5% (95%CI 1.0%, 9.0%) [32]. As this review was conducted in 2020, there were still no studies to assess the frequency differences according to the second version of the EWGSOP and AGWS criteria. Another more recent systematic review describes a prevalence of 22% (95%CI 20.0%, 25.0%) of sarcopenia in adults assessed by the EWGSOP1 criterion. In this systematic review, the frequency assessed by the EWGSOP2 criterion was 10.0% (95%CI 20.0%, 25.0%), while the frequencies by AGWS1 and IWGS were similar [15% (95%CI 13.0% and 17.0%) and 14% (95%CI 9.0%, 18.0%), respectively] [33]. In our study, the frequency of sarcopenia in people with HF assessed by the EWGSOP2 criteria was 23.0% and was lower than that assessed by the EWGSOP1 criteria (31%). However, we found higher frequencies when the AGWS1 criteria were used (26%).

Other systematic reviews have assessed the frequency of sarcopenia in people with HF [36,37]. However, most of them included patients admitted to the hospital, which may affect the estimation of the frequency of sarcopenia in this population [36]. It is believed that acute illness may affect muscle mass [12]. Zhang et al. found a frequency of sarcopenia of 36% (95%CI 26%, 47%) in people with HF [37]. This prevalence was higher than that found in our study, but the inclusion of studies with inpatients in the meta-analysis by Zhang et al. may explain this difference. When these authors performed subgroup analysis, the frequency of sarcopenia was higher in inpatients when compared to outpatients [55.0% (95%CI 43.0%, 66.0%) vs. 26.0% (95%CI 16.0%, 37.0%)] [37].

In our study, the method of muscle measurement and the procedure of reporting muscle mass were not associated with the frequency of sarcopenia in the meta-regression. This lack of association may have occurred because both methods are sensitive to the amount of body water [38]. Factors such as changes in hydration status and edema can interfere with measuring muscle mass [38]. These factors can affect the interpretation of both bioimpedance and DXA. Other factors, such as food intake and exercise before the exam, can affect bioimpedance [38]. Conversely, DXA can be affected by the amount of intramuscular adipose tissue [38]. However, more accurate methods, such as computed tomography (CT) and nuclear magnetic resonance (MRI), are expensive and difficult to access in clinical practice [12,35]. Besides, neither CT nor MRI have well-defined cohort points for diagnosing sarcopenia [12,38]. Thus, bioimpedance and DXA are still the preferred methods for diagnosing sarcopenia in clinical practice.

Several factors present in people with HF may contribute to a higher risk of sarcopenia. Among them are described neurohormonal hyperstimulation, changes in GH and IGF1, decreased testosterone, insulin resistance, vitamin D deficiency, chronic inflammation (with changes in TNF-alpha, C-reactive protein, interleukin 6), increased oxidative stress, mitochondrial dysfunction, malabsorption secondary to edema, anorexia related to HF itself and the medications used to treat it, low protein intake, loss of ions caused by the use of diuretics, polypharmacy, low

mobility, sedentary lifestyle and smoking [39,40]. These factors may contribute to and accentuate muscle dysfunction in this population.

The WHO considers Chagas disease a tropical disease with strong social determinants and is often neglected [5]. This disease is considered endemic in several Latin American countries and has been reported in 44 countries, including the United States, Canada, European countries, Western Pacific countries, and Eastern Mediterranean countries [5]. Despite its wide distribution, studies on this disease and its manifestations still need to be made [4,5]. In our systematic review, we found only few studies that evaluated the frequency of Sarcopenia in HF-C [19,20,24]. In these studies, the pooled frequency of sarcopenia was 24.2%. Some characteristics present in Chagas, such as systemic infection by the parasite with persistent chronic inflammation, dysautonomia, microvascular disturbances, and immune-mediated myocardial injury [2–4,41], may contribute to the high prevalence of sarcopenia in HF-C.

Our study has some limitations. Our systematic review mainly included observational studies. For this reason, some heterogeneity is expected. Nevertheless, the heterogeneity in the frequency of sarcopenia was partially explained by the NYHA class and ejection fraction. Besides, other factors not evaluated by most studies, such as medication use, duration of heart failure, and causes of heart failure, may have contributed to heterogeneity. Another limitation of our systematic review was the lack of comparison and definitions of cachexia. It is possible that, in some studies, these two entities overlap. We attempted to minimize this problem by including only outpatients. Finally, the results on sarcopenia in HF-C should be interpreted cautiously due to the paucity of studies. However, the results of our systematic review are crucial for mapping the research gap on PHFC and highlighting the need for further studies to assess sarcopenia in this population.

Our systematic review also has some strengths. Most of the included studies were of high quality and assessed using the NOS, which contributed to the reliability of our results. Furthermore, we were able to map the frequency of sarcopenia according to the different operational diagnostic criteria, which may contribute to the choice of criteria for heart failure and the translation of research findings into clinical practice.

In conclusion, our systematic review indicates that people with HF are at increased risk for sarcopenia. Further, we found that people with HF are 2.35 times more likely to present Sarcopenia when compared to people without HF. These findings have potential implications for the quality of life of patients with HF and should be considered in the care provided to these individuals. Our findings highlight the hypothesis that sarcopenia could be a problem in HF-C. Nevertheless, more studies are needed to evaluate sarcopenia in HF-C.

## Supporting information

**S1 Text. Supplementary files for tables and figures.**
(DOCX)

**S1 Checklist. PRISMA 2020 Checklist [11].**
(DOCX)

## Author contributions

**Conceptualization:** Melissa Orlandin Premaor, Suellen de Azevedo Mendes, Gustavo Henrique Silva Ambrósio Vieira, Camila Diniz Braga, Aron Fonseca Santos, Diego Silva Assaf Ferreira, Bruno Cesar Fernandes Araujo, Maria do Carmo Pereira Nunes, Luis Felipe Jose Ravic de Miranda.

**Data curation:** Melissa Orlandin Premaor, Luis Felipe Jose Ravic de Miranda.

**Formal analysis:** Melissa Orlandin Premaor, Gustavo Henrique Silva Ambrósio Vieira, Camila Diniz Braga, Aron Fonseca Santos, Diego Silva Assaf Ferreira, Bruno Cesar Fernandes Araujo, Maria do Carmo Pereira Nunes, Luis Felipe Jose Ravic de Miranda.

**Funding acquisition:** Melissa Orlandin Premaor.

**Investigation:** Melissa Orlandin Premaor, Suellen de Azevedo Mendes, Gustavo Henrique Silva Ambrósio Vieira, Camila Diniz Braga, Aron Fonseca Santos, Diego Silva Assaf Ferreira, Bruno Cesar Fernandes Araujo, Maria do Carmo Pereira Nunes, Luis Felipe Jose Ravic de Miranda.

**Methodology:** Melissa Orlandin Premaor, Gustavo Henrique Silva Ambrósio Vieira, Maria do Carmo Pereira Nunes, Luis Felipe Jose Ravic de Miranda.

**Project administration:** Melissa Orlandin Premaor, Luis Felipe Jose Ravic de Miranda.

**Resources:** Luis Felipe Jose Ravic de Miranda.

**Supervision:** Melissa Orlandin Premaor, Maria do Carmo Pereira Nunes, Luis Felipe Jose Ravic de Miranda.

**Validation:** Melissa Orlandin Premaor, Maria do Carmo Pereira Nunes, Luis Felipe Jose Ravic de Miranda.

**Visualization:** Melissa Orlandin Premaor, Maria do Carmo Pereira Nunes, Luis Felipe Jose Ravic de Miranda.

**Writing – original draft:** Melissa Orlandin Premaor, Suellen de Azevedo Mendes, Camila Diniz Braga, Aron Fonseca Santos, Diego Silva Assaf Ferreira, Bruno Cesar Fernandes Araujo, Luis Felipe Jose Ravic de Miranda.

**Writing – review & editing:** Melissa Orlandin Premaor, Maria do Carmo Pereira Nunes, Luis Felipe Jose Ravic de Miranda.

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
