## [Decision Letter · Decision Letter 0]

5 Aug 2025

PEOPLE WITH HEART FAILURE, SARCOPENIA AND CHAGAS DISEASE: A SYSTEMATIC REVIEW AND META-ANALYSIS

Dear Dr. Premaor,

Thank you for submitting your manuscript to PLOS Neglected Tropical Diseases. After careful consideration, we feel that it has merit but does not fully meet PLOS Neglected Tropical Diseases's publication criteria as it currently stands. Therefore, we invite you to submit a revised version of the manuscript that addresses the points raised during the review process.

Please submit your revised manuscript within 60 days Oct 04 2025 11:59PM. If you will need more time than this to complete your revisions, please reply to this message or contact the journal office at plosntds@plos.org. Please include the following items when submitting your revised manuscript:

We look forward to receiving your revised manuscript.

Kind regards,

Andrés F. Henao-Martínez, M.D.

Academic Editor

Susan Madison-Antenucci

Section Editor

Shaden Kamhawi

co-Editor-in-Chief

Paul Brindley

co-Editor-in-Chief

**Journal Requirements:**

1) Please provide an Author Summary. This should appear in your manuscript between the Abstract (if applicable) and the Introduction, and should be 150-200 words long. The aim should be to make your findings accessible to a wide audience that includes both scientists and non-scientists. Sample summaries can be found on our website under Submission Guidelines:

4) Please ensure that the funders and grant numbers match between the Financial Disclosure field and the Funding Information tab in your submission form. Note that the funders must be provided in the same order in both places as well.

**Reviewers' Comments:**

Reviewer's Responses to Questions

**Key Review Criteria Required for Acceptance?**

**Methods**

-Are the objectives of the study clearly articulated with a clear testable hypothesis stated?

-Is the study design appropriate to address the stated objectives?

-Is the population clearly described and appropriate for the hypothesis being tested?

-Is the sample size sufficient to ensure adequate power to address the hypothesis being tested?

-Were correct statistical analysis used to support conclusions?

-Are there concerns about ethical or regulatory requirements being met?

Reviewer #1: In the study design, mention that it is also a meta-analysis.

Describe acronym meaning PRISMA- Preferred Reporting Items for Systematic Reviews and Metanalyses (PRISMA).

If the aim is to assess sarcopenia and HF, it is not understandable to exclude hospitalized patients.

“The previously defined terms for each database are described in the supplementary material” . number the material in the text for easy identification, and keep it consistent with the citation in the text

Reviewer #2: The objective and hypothesis are well-described. A systematic review is appropriate to summarize evidence. The selected papers met the pre-established inclusion/exclusion criteria based on the clinical definition. However, despite the 25 reviewed articles included >2500 heart failure patients, the subgroup of Chagas disease (CD) patients is underpowered (only 2-3 studies assessed sarcopenia in CD patients), limiting conclusive results for the proposed comparison. No concern about ethical or regulatory requirements.

Reviewer #3: The study was conducted in accordance with PRISMA guidelines, and its protocol was registered in PROSPERO, reflecting a commitment to transparency and methodological rigor. Key aspects of the analysis—such as eligibility criteria, data sources, and statistical methods—are aligned with best practices for systematic reviews and meta-analyses. The authors applied appropriate inclusion and exclusion criteria, focusing on adult patients with heart failure (HF) of any etiology, including Chagas disease, while explicitly excluding studies involving specific subgroups, thereby reducing the risk of selection bias. The methods section demonstrates a structured and comprehensive approach to study selection, data extraction, and quality assessment.

Reviewer #4: (No Response)

**Results**

-Does the analysis presented match the analysis plan?

-Are the results clearly and completely presented?

-Are the figures (Tables, Images) of sufficient quality for clarity?

Reviewer #1: Studies that evaluated the same population should be excluded as they may overestimate the data, including prevalence.

Perhaps the ideal would be to include only the additional information from the cohort, and not the prevalence.

How did you manage to avoid this bias?

Cvjetan et al.[14] is an update from the conference paper by Loncar et al.[15

Fonseca et al.[16], Fonseca et al.[17, 18], Emmani et al.[19], and dos Santos et al.[20] included subjects from the same cohort.

In table 2 there is a blank row

"One study has estimated muscle mass using the Forbes formula”. reference the study.

Was it excluded from the data analysis or review? If it was excluded from the review, it would be interesting to describe it in the flow of included studies

Reviewer #2: Multiple diagnostic criteria for sarcopenia were included, which are not directly comparable. Despite of this, the

the statistical analysis was correct, including heterogeneity assessment. The pooled OR for CH patients with sarcopenia was not statistically significant, with a wide confidence interval, indicating high uncertainty. The exclusion of hospitalized patients, aimed to reduce confounding from acute illness, also limits the generalizability.

Reviewer #3: The results presented are fully consistent with the analysis plan described in the methods section. All proposed procedures were executed. The presentation of the results is clear, structured, and comprehensive. The description details the selection processes, characteristics of the included studies, methods for assessing muscle mass and sarcopenia, and the comparison of sarcopenia prevalence between patients with and without heart failure, as well as between patients with Chagas disease-related heart failure and other etiologies.

Reviewer #4: I believe the authors should consider moving some of the topics currently presented in the supplementary material into the main body of the manuscript, in order to facilitate readers’ analysis and comprehension.

**Conclusions**

-Are the conclusions supported by the data presented?

-Are the limitations of analysis clearly described?

-Do the authors discuss how these data can be helpful to advance our understanding of the topic under study?

-Is public health relevance addressed?

Reviewer #1: The confidence intervals for the odds ratio are similar between IC and Chagas-related IC. So I believe they can be considered similar, rather than higher. Moreover, the prevalence was slightly higher and was assessed in only two studies.

Reviewer #2: The conclusion that the risk of sarcopenia in HF Chagas patients might be higher than HF from other causes is not supported by the data, given the small number of patients and the wide OR. A more cautious statement should mention that the results should be a hypothesis-generating observation, not a finding. Also, it is important to acknowledge the geographical limitations of the data. The authors acknowledged that co-morbidities, frailty, chronic disease, etc, are relevant confounders and may overlap with sarcopenia, but this was not addressed in the data. The inconclusive results limit its public health relevance at this point.

Reviewer #3: This systematic review presents consistent evidence that patients with heart failure—including those with Chagas disease—are at increased risk of developing sarcopenia, particularly when assessed using recent and more stringent diagnostic criteria. While the findings are limited by heterogeneity among studies and the relatively small number of investigations focused specifically on Chagas disease, this review provides important contributions to the current understanding of sarcopenia in neglected populations and identifies clear priorities for future research.

Reviewer #4: The manuscript is well-structured and clearly written. Although the number of referenced studies is limited, the work remains methodologically sound and scientifically relevant. The authors are encouraged to continue advancing research in this field, particularly given the importance of understanding sarcopenia in the context of heart failure and neglected diseases such as Chagas disease.

**Editorial and Data Presentation Modifications?**

Reviewer #1: “Among those infected, approximately 30% develop chronic cardiomyopathy with arrhythmias and heart failure”. • There is a recent systematic review that reports this percentage as 42%

Reviewer #2: Clarify and temper the conclusions, which are just speculative. Attempt to stratify results by diagnostic criteria and sarcopenia definitions. A supplementary table could present sarcopenia criteria in each study to facilitate readers. Standardize terminology across the manuscript and review grammar. The discussion and methods sections could be more concise.

Reviewer #3: (No Response)

Reviewer #4: 1. Could the authors standardize the presentation of the results throughout the manuscript, particularly between the abstract and the results section?

Harmonizing the reporting format would enhance clarity and comparability of the findings.

2. Could the authors provide a more in-depth explanation of the source of heterogeneity observed in the main meta-analysis, especially considering the I² value of 77.2% in the comparison between individuals with heart failure and the control group?

3. Could the authors elaborate further on the clinical and epidemiological implications of sarcopenia prevalence in individuals with heart failure due to Chagas disease, particularly in light of the scarcity of data in Latin American populations and the known impact of sarcopenia on morbidity and mortality?

4. Although the leave-one-out analysis was appropriately performed, could the authors offer a clinical interpretation of the influence of studies such as Formiga et al. and Karim et al. on the observed heterogeneity?

**Summary and General Comments**

Reviewer #1: (No Response)

Reviewer #2: This systematic review addresses an important and under-researched topic. The study contributes novel insights into sarcopenia in a neglected population. It highlights a need for tailored clinical approaches in Chagas-endemic regions.

The authors should be commended for their effort to synthesize the available evidence across multiple studies and apply a structured methodology aligned with PRISMA and GRADE guidelines. The registration of the review protocol on PROSPERO and the inclusion of a meta-regression analysis reflect good scientific rigor. However, the conclusions overreach the presented data, and the methodological heterogeneity significantly limits the strength of the findings. The overstated conclusions are drawn from small and statistically inconclusive subgroups.

Reviewer #3: (No Response)

Reviewer #4: (No Response)

PLOS authors have the option to publish the peer review history of their article (what does this mean? ). If published, this will include your full peer review and any attached files.

**Do you want your identity to be public for this peer review?** For information about this choice, including consent withdrawal, please see our Privacy Policy .

Reviewer #1: No

Reviewer #2: **Yes: ** Fabio Zicker

Reviewer #3: **Yes: ** Elaine Cristina Navarro

Reviewer #4: **Yes: ** Tycha Bianca Sabaini Pavan

**Figure resubmission:**

**Reproducibility:**



---

## [Decision Letter · Decision Letter 1]

29 Oct 2025

Dear Dr. Premaor,

We are pleased to inform you that your manuscript 'PEOPLE WITH HEART FAILURE, SARCOPENIA AND CHAGAS DISEASE: A SYSTEMATIC REVIEW AND META-ANALYSIS' has been provisionally accepted for publication in PLOS Neglected Tropical Diseases.

Best regards,

Susan Madison-Antenucci

Section Editor

Shaden Kamhawi

co-Editor-in-Chief

Paul Brindley

co-Editor-in-Chief

Reviewer's Responses to Questions

**Key Review Criteria Required for Acceptance?**

**Methods**

-Are the objectives of the study clearly articulated with a clear testable hypothesis stated?

-Is the study design appropriate to address the stated objectives?

-Is the population clearly described and appropriate for the hypothesis being tested?

-Is the sample size sufficient to ensure adequate power to address the hypothesis being tested?

-Were correct statistical analysis used to support conclusions?

-Are there concerns about ethical or regulatory requirements being met?

Reviewer #1: (No Response)

Reviewer #2: This is a revised version of the manuscript. Our comments were adequately replied/addressed, and the text has been amended appropriately. The authors acknowledged limitations of the study. I have no further questions.

Reviewer #4: (No Response)

**Results**

-Does the analysis presented match the analysis plan?

-Are the results clearly and completely presented?

-Are the figures (Tables, Images) of sufficient quality for clarity?

Reviewer #1: (No Response)

Reviewer #2: Yes.

Reviewer #4: (No Response)

**Conclusions**

-Are the conclusions supported by the data presented?

-Are the limitations of analysis clearly described?

-Do the authors discuss how these data can be helpful to advance our understanding of the topic under study?

-Is public health relevance addressed?

Reviewer #1: (No Response)

Reviewer #2: The conclusions, as well as the limitations of data and analysis, have been acknowledged by the authors and are reflected in the revised version.

Reviewer #4: (No Response)

**Editorial and Data Presentation Modifications?**

Reviewer #1: (No Response)

Reviewer #2: I have no further comments/suggestions

Reviewer #4: It is requested that an additional term or expression be included to reinforce that the referenced articles are already encompassed within the previously defined universe of 25 publications. Although this may introduce a degree of redundancy, such an approach enhances clarity and interpretative accuracy for the reader.

In addition to reporting the minimum and maximum age values, it is recommended to present the mean, as well as the first and third quartiles (Q1 and Q3).

**Summary and General Comments**

Reviewer #1: (No Response)

Reviewer #2: Strengths, weaknesses, relevance, novelty, significance, and execution were addressed in the initial review. The revised version satisfactorily addresses this reviewer's questions.

Reviewer #4: (No Response)

PLOS authors have the option to publish the peer review history of their article (what does this mean? ). If published, this will include your full peer review and any attached files.

**Do you want your identity to be public for this peer review?** For information about this choice, including consent withdrawal, please see our Privacy Policy .

Reviewer #1: **Yes: ** Ariela Mota Ferreira

Reviewer #2: **Yes: ** Fabio Zicker

Reviewer #4: **Yes: ** Tycha Bianca Sabaini Pavan

---

## [Editor Report · Acceptance letter]

Dear Dr. Premaor,

We are delighted to inform you that your manuscript, "PEOPLE WITH HEART FAILURE, SARCOPENIA AND CHAGAS DISEASE: A SYSTEMATIC REVIEW AND META-ANALYSIS," has been formally accepted for publication in PLOS Neglected Tropical Diseases.

Best regards,

Shaden Kamhawi

co-Editor-in-Chief

Paul Brindley

co-Editor-in-Chief
